# Structural balance emerges and explains performance in risky decision-making

Omid Askarisichani [1], Jacqueline Ng Lane[2], Francesco Bullo [3,4], Noah E. Friedkin [3,5], Ambuj K. Singh[1] & Brian Uzzi[6,7]

Polarization affects many forms of social organization. A key issue focuses on which affective relationships are prone to change and how their change relates to performance. In this study, we analyze a financial institutional over a two-year period that employed 66 day traders, focusing on links between changes in affective relations and trading performance. Traders' affective relations were inferred from their IMs (>2 million messages) and trading performance was measured from profit and loss statements (>1 million trades). Here, we find that triads of relationships, the building blocks of larger social structures, have a propensity towards affective balance, but one unbalanced configuration resists change. Further, balance is positively related to performance. Traders with balanced networks have the "hot hand", showing streaks of high performance. Research implications focus on how changes in polarization relate to performance and polarized states can depolarize.

[1] Department of Computer Science, University of California, Santa Barbara, CA 93106, USA. [2] Harvard Business School, Harvard University, Boston, MA 02134, USA. [3] Center for Control, Dynamical Systems and Computation, University of California, Santa Barbara, CA 93106, USA. [4] Department of Mechanical Engineering, University of California, Santa Barbara, CA 93106, USA. [5] Department of Sociology, University of California, Santa Barbara, CA 93106, USA. [6] Northwestern Institute on Complex Systems, Northwestern University, Evanston, IL 60208, USA. [7] Management and Organizations Department, Northwestern University, Evanston, IL 60208, USA. Correspondence and requests for materials should be addressed to B.U. (email: uzzi@kellogg.northwestern.edu)

Recent world events have rekindled interest in social networks of positive and negative relations. Examples prevail across geopolitics, settings where firms compete on new standards of innovation, national elections, social media, religious groups, and many other situations where polarization is frequent. Despite the many real-world settings where interpersonal riffs among collaborators can arise and potentially undermine performance[1–4], research on how positive and negative relationships among collaborators change and how those changes relate to performance is relatively nascent[5–7]. Newly available data on the electronic communications among networks of individuals enable an opportunity to measure changes in interpersonal sentiments and their relationship with changes in performance of the system[8,9].

Structural balance theory (SBT) provides an analytical framework for measuring and predicting how polarized sentiments among collaborators change and relate to performance. SBT characterizes every individual relationship as being either positive or negative in sentiment and is classically defined on directed networks[10–13]. Positive sentiments include ally, friend, or supporter relationships and negative sentiments include competitor, foes, or detractor relationships. On the basis of four rules of interaction, SBT posits whether relationships will remain polarized (unbalanced) or will reconfigure, i.e., become "balanced". The four rules are: a friend of a friend is a friend, a friend of an enemy is an enemy, an enemy of an enemy is a friend, and an enemy of a friend is an enemy. These four rules disaggregate a network of ties into 16 different types of triads of relationships. Triads can be characterized as balanced or polarized. Two of the 16 feasible triads are considered structurally balanced and balanced configurations have a propensity for stability. Polarized configurations are prone to dissolution and reorganization. Aggregating local triads provides non-intuitive implications for a group's macrostructure. A group's network topology moves towards either a complete network of all-positive sentiments or a network partitioned into two subgroups with no negative within-group sentiments and all negative between-group sentiments. Implicit in SBT is that stable configurations should support higher performance than polarized, unstable configurations. Thus, by examining micro patterns of sentiment changes, SBT enables understanding of how interpersonal relationships evolve and how these configurations either enable or hinder performance.

A sequence of generalizations followed, reviewed in ref. [12], toward a SBT model in which nine of 16 triad types are permissible and the remainder set of seven are forbidden based on one or more violations of transitivity (if A likes B, and B likes C, then A likes C) in a triad's configuration of sentiments. This line of advancement was associated with empirical investigations of networks in field-settings as in ref. [14], which evaluated whether the distribution of observed triads over the 16 feasible types indicated a bias toward a set of SBT model-specific permitted triads. The current frontier of work on SBT is focused on modeling advancements of the temporal evolution of sentiment networks[15–21]. These temporal models are motivated by the idea that field-setting networks are undergoing transformations in which positive sentiments are being converted to negative sentiments, and vice versa, toward the attractor state of structural balance. Investigations of longitudinal data on sentiment networks in field-settings, relevant to these dynamical models, are rare[16,22,23]. Moreover, despite evidence that social networks affect performance in task-oriented groups[24], there have been limited opportunities to examine the effects of structural changes on performance over time. This article reports findings from the most extensive set of longitudinal data yet assembled to evaluate the theory's prediction of an evolution toward structural balance,

and to investigate whether sentiment network states are linked with changing task performance metrics. Our investigation draws on a unique dataset from a financial trading firm to test dynamic predictions and to evaluate whether sentiment network states are linked with task performances in a competitive risky decision-making environment.

First, we find a tendency for the sentiment network to steadily transition into states of greater balance over time, that is, with toward fewer violations of SBT predictions than expected in a suitability randomized network. Second, using Markov Chain analysis, we find that only certain types of triads tend to transition from states that violate SBT predictions to states with no violations. Third, we find that an individual trader's degree of structural balance is positively associated with the trader's performance. There is temporal evidence that structural balance and performance are mutually reinforcing. Trader performance increases as the degree of a trader's embedding in classical balanced triads increases, after accounting for individual trader differences and market uncertainty.

## Results

**Trading firm network**. We analyzed the starting, developmental, and ending states of the sentiment network of a medium-sized trading firm over a 2-year period. A trading firm employs stock traders who invest the firm's money in the stock market with the expectation of maximizing the firm's return on invested capital. Day traders typically open new positions each day, trade those positions during the day, and then sell off all holdings by the end of the day. Consequently, a day trader's performance is measured on a day-to-day basis. Relationships in the firm are flat and non-hierarchical. All traders are at the same administrative level/rank and have relative autonomy in choosing the stocks they trade within the constraints of making money for the firm. Traders voluntarily form attachments with other traders to gain information relevant to their trading performance[25–31]. Typically, because relationships affect a trader's performance and create opportunities to celebrate victories and commiserate losses, traders with ongoing attachments trust and like one another[25,32–34].

To measure relationships among traders, we analyzed 128,323 instant messages, including content, as well as 14,259 trades of the dynamic sentiment network of stock traders in the firm from October 2007 to March 2009[35]. We extracted all social messages from the instant messages using content analysis because they are indicators of individuals' interpersonal, rather than instrumental relationships. On average, traders sent $228.82 \pm 40.22$ IM's per quarter to $5.98 \pm 0.48$ contacts, with a closeness centrality score of $0.15 \pm 0.04$. The network had an average clustering coefficient of $0.35 \pm 0.04$.

The complete record of IM exchanges and trades provides empirical advantages over prior work, including (i) a novel application of SBT to utilitarian relationships, in contrast to pure friendships[16,22,36], (ii) a minimization of self-report and mono-method biases[37], and (iii) extensive high resolution longitudinal data. All data are taken directly from the firm's servers, which archive all communication and trading data per SEC regulations. The Institutional Review Board of Northwestern University approved the study (See Methods for data and measurement details).

**Structural balance triads**. Table 1 describes the 16 triad types. We use the classical SBT model definition of structural balance and operationalizations of positive and negative edges (see Methods for details). Its four axioms are: (A1) A friend of a friend is a friend, (A2) A friend of an enemy is an enemy, (A3) An enemy of a friend is an enemy, and (A4) An enemy of an enemy

**Table 1 SBT's 16 types of triads**

| Triad Type | Triad Label | A1 | A2 | A3 | A4 |
|---|---|---|---|---|---|
|  | 300 | | | | |
|  | 102 | | | | |
|  | 003 | | | | X |
|  | 120D | | X | | |
|  | 120U | | | X | |
|  | 030T | | X | X | X |
|  | 021D | | | X | X |
|  | 021U | | X | | X |
|  | 012 | | | | X |
|  | 021C | X | | | X |
|  | 111U | X | | X | |
|  | 111D | X | X | | |
|  | 030C | X | | X | |
|  | 201 | X | X | X | |
|  | 120C | X | X | X | |
|  | 210 | X | X | X | |

Triads have six positive or negative sentiments (only positive sentiments are displayed) and are characterized by three numbers: the number of mutual (M), asymmetric (A), and null (N) ties, and symbols that discriminate triads with identical MAN numbers – transitive (T), up (U), down (D), and cyclic (C)

Table 1. Here we show the steps involved in computing the transition probability from the unbalanced or polarized triad state 210 to the balanced triad state 300 over time period $(t, t + 1)$.

**Markov transitions**. Figure 2 shows the Markov transition probability matrix for each quarterly period of the likelihood of transition between any two triadic configurations states. Each row represents a transition out of a state $i$ and each column represents a transition into a state $j$; stability of a state is represented by the diagonal (see Methods for details). For example, the propensity for the non-balanced triad number 210 in the last row to transition to the triad number 300 in the first row of all matrices is ~0.3. The transition probabilities highlight three important insights and demonstrate support for the tenets of SBT in dynamically measured settings.

First, the Markov transition probabilities are relatively stable across transition periods, as indicated by the high degree of similarity between the triad count ratios, $c_{xt}$, for each state and their corresponding stationary distributions. This is supported by the low L2-Norm distances of stationary probability distribution from the average distribution and steady stability ratios in the subsequent periods (Table 2), where the stability ratio denotes the proportion of transitions in the observed transition matrix that were statistically significant compared to the randomized transition matrices for each period. The stability ratios for each transition period indicate that traders reconfigure attachments in a consistent manner over time such that the overall system maintains a stable transition probability distribution and that the observed transitions are not likely to be explained by chance (Table 2). See Method for methodological details and measurement robustness checks.

Second, examining the final stationary probability distributions associated with each triad configuration (Table 3), we find that the probability associated with being in one of the remaining 13 unbalanced states, excluding the null triad (Table 1), is just 0.03. This compares to a 0.22 probability of being in one of the two classical balanced states (and a ~0.97 probability of being in balanced states allowed based on Davis et al.[38]). In this study, we only use Davis' theoretical deductions from his formal model. Also, we find the distributions of triads are consistently a close match to the stationary distribution over periods. Therefore, the system has very low occurrences of unbalanced states (i.e., near zero) at each period of analysis and is consistent with SBT's predictions by Heider et al. and Davis et al.[38,39].

Third, we observe a strong propensity for stability in the "classic" balanced states, 300 and 102, as well as the null triad state, 003 (Table 1), indicating that the trader network has a tendency towards clustering into two or more subgroups[38]. Heider predicted this finding in his seminal work[39]. In particular, Heider writes "if two negative relations are given, balance can be obtained either when the triad relationship is positive or when it is negative, though there appears to be a preference for the positive alternative"[38,39]. Davis subsequently introduced the formal theoretical model, which he called "clustering"[38] that allows for the triad 003. Finally, the overlooked prediction by Heider and Davis in balance theory[38,39], turns out to have empirical support in a longitudinal field setting. Further, it suggests that once a triad enters the states of 300 or 102, it has a low probability of transitioning out of its current state. Thus, once traders have reconfigured their ties to a state of structural balance, they remain in these balanced configurations. Similarly, the stationarity of the null triad state suggests that the network of positive attachments remains relatively sparse over time.

is a friend. The more general terms "positive" and "negative" relationships (sentiments) are often substituted for the metaphorical terms "friend" and "enemy" in practice. Thus, each triad entails six positive or negative sentiments. Only positive sentiments are displayed. A triad type with at least one violation of these axioms is a "forbidden" triad. It can be shown that in a sentiment network with no violations of any of these four axioms, only two types of triads may exist: 300 and 102. We refer to these two types as "permitted" triads.

Figure 1 conceptualizes how Markov Chain analysis is used to compute the state transition probabilities for the 16 triad types in

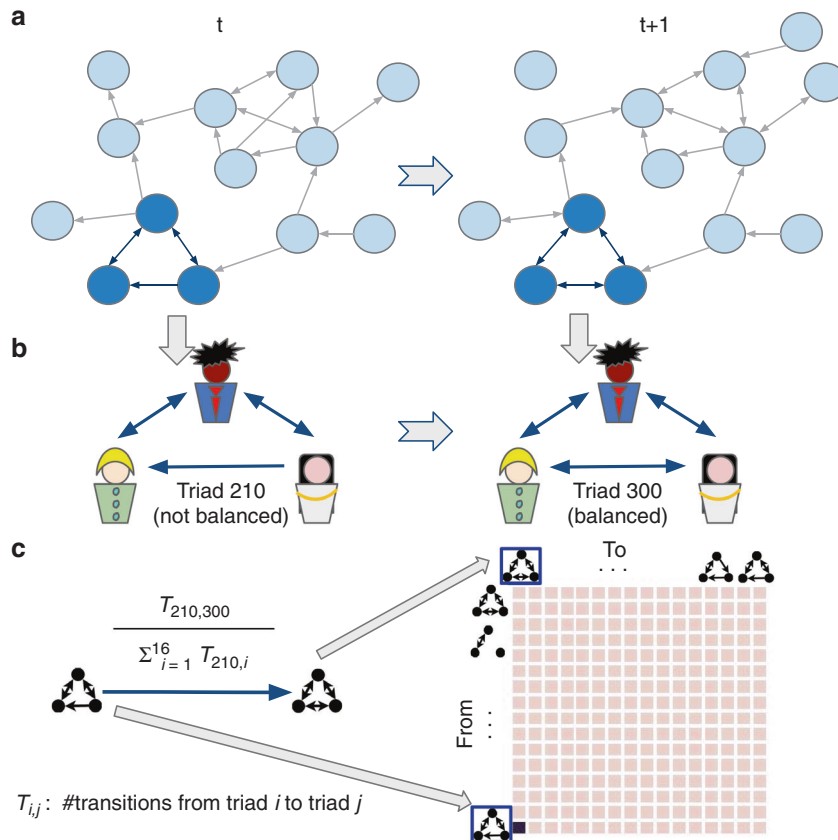

**Fig. 1** Illustrative figure showing state transition probabilities from unbalanced or polarized triad state (210) to balanced triad 300. **a** For each period, we extract a directed graph of social IM's among traders, and identify interpersonal relations by comparing the observed relations against a statistical null-model based on Wuchty et al.[35]. **b** We compute transition probabilities between periods for each observed triad. In this example, we demonstrate the configuration of sentiments for three illustrative nodes and compute the corresponding Markov transition probability from triad 210–300. **c** We repeat for each triad in each period, resulting in a 16 state (triad) Markov Chain capturing the complete transition probabilities between states and periods (See Methods)

**Balance in randomized networks**. To test whether the observed triad states can be explained by chance interactions among the traders, we compare the likelihood of observing each triad relative to the corresponding triad in 10,000 suitably randomized networks (See Methods), for each of the 6 time periods, shown in Fig. 3. Informed by the stationary probability distributions (Table 3) of particular interest is the likelihood of observing the classical balanced (i.e., 102 and 300) and null (i.e., 003) triads in the actual network compared to the randomized networks. Examining Fig. 3, we find that while both balanced triad states are significantly more likely to occur in the actual network compared to the randomized network, our actual network has a lower occurrence of null triads than a randomized network would suggest. The figure is computed unrelated to transition probabilities, yet shows the high significance of balanced triads. Accordingly, Fig. 3 confirms that the underlying assembly rules of balance theory influence the reconfiguration of interpersonal sentiments in the network towards increased balance, beyond what a random network would imply.

Further, Fig. 4 compares the observed degree of classic balance over time $b_t$ to the expected $\hat{b}_t$ of the randomized network and validates that the observed, $b_t$ is significantly higher than the expected $\hat{b}_t$ derived from the randomized network, for all observed time periods. Actual networks consistently showed significantly higher balance than the randomized networks. This finding shows that the observed triad states are not explained by chance interactions. That said, we find that the overall ratio of classically balanced triads decreases over time. This decline corresponds to the 2008–2009 financial crash and aligns with prior work suggesting that a communication network tends to "turtle up" during periods of uncertainty[34].

Thus, the relative likelihood of occurrence of the remaining unbalanced states in our observed network, while small, display significant differences between the structures in the network and those of a randomized network. These structural differences reflect the underlying dynamics of our particular context, as well as the social norms associated with instant messaging communication. However, despite these noted discrepancies, the stationary probability distributions (Table 3) confirm that the unbalanced configurations occur with very low probabilities and do not detract from the overall trend towards structural balance in the system. Notably, although unbalanced triads are moving towards greater balance, these transitions occur slowly; hence, few forbidden triads (201 and 021) remain within our observation period (Table 3).

An untested premise of SBT is that balance positively relates to performance[40]. Existing research indicates that ceteris paribus persons choose professional attachments they like and trust ("lovable fools") over more skillful attachments ("competent jerks") because ongoing attachments create lock-ins that lead persons to value the good relationships over performance[5,32,41].

**Balance and performance**. We investigated the untested link between structural balance and trader performance by regressing an individual's trading performance on their balance $b_{it}$. Balance

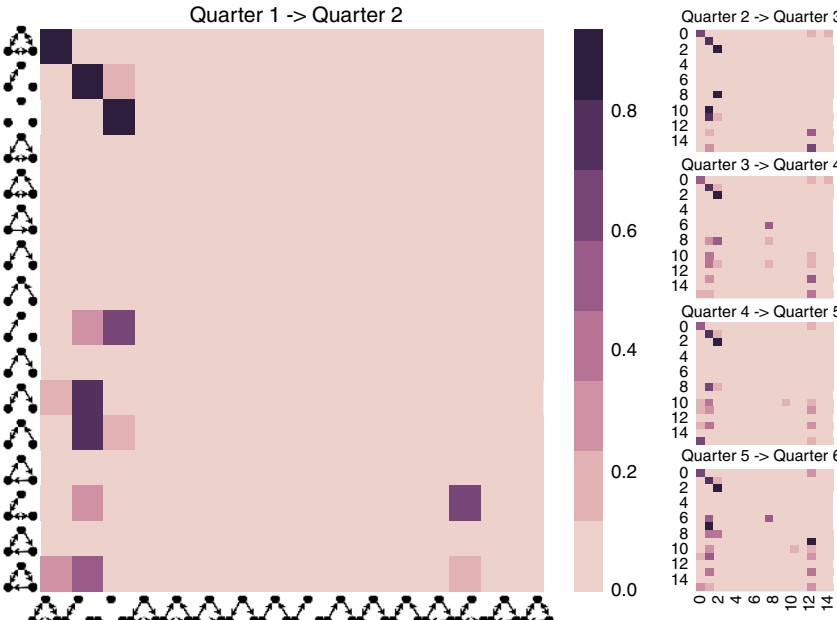

**Fig. 2** Stochastic Markov transition matrices of observing a given transition, $p_{ij}(t)$ over the period $(t, t+1)$ for all traders. Row values correspond to transitions out of a triad, column values correspond to transitions into a triad, and diagonals correspond to triad stability probabilities. Probabilities are stable across different periods and different threshold-based methodologies. Transitions occur from unbalanced to balanced triads but not vice versa. The presence of such transitions suggests once traders have reconfigured their ties to a state of structural balance, they remain in these balanced configurations

**Table 2 Stability of transition probabilities: L2-Norm distance of the stationary probability distribution from their average is relatively stable**

| Transition period | L2-Norm distance | Stability ratio of randomized networks |
|---|---|---|
| 1–2 | 0.07 | 0.84 |
| 2–3 | 0.08 | 0.92 |
| 3–4 | 0.06 | 0.94 |
| 4–5 | 0.05 | 0.91 |
| 5–6 | 0.20 | 0.83 |

Each state is defined as a vector of 16 probabilities. Stability test shows that at least 83% of transitions in each observed Markov chains are statistically significant compared to the ones computed from randomized networks

of trader $i$ at period $t$ is trader $i$'s ratio of classically balanced triads (i.e., configuration 102 or 300 in Table 1) to total triad configurations in period $t$. Individual monthly performance, $profit_{it}$ was assessed by measuring whether trader $i$ does better or worse than their mean individual-level performance across all time periods, i.e., whether a trader's structural balance is related to getting a "hot hand" in the market[42]. We use monthly performance because unlike the first set of analyses examining structural balance, where our focus was the long-term reconfigurations of interpersonal relationships, our focus here is on the near-term implications of balance on day traders' performance. Formally, our outcome variable is whether trader $i$ performs better ($profit_{it} > \frac{1}{N} \sum_{t'=1}^{N} profit_{it'}$) or worse than their individual-level mean profit across all periods, where $N$ is the number of periods. This variable is coded as $p_{it} = 1$ or $p_{it} = 0$, respectively. In our regression models we control for other factors influencing trading success, including market volatility (1 = high, using the standard measure of the VIX), trader fixed effects, period fixed effects, average trade value ($), active trading days, trader's degree centrality and IM's sent. Trader balance is measured as the log of

balance. A $\text{Logit}(p_{it}) = \beta_0 + \beta_1(b_{it})$ regression was used to test the relationship and further validated with a non-parametric regression. The non-parametric regression imposes no distributional assumptions on the data or misspecification errors and provides a stringent test of the hypothesis by using 10-fold cross validation and bootstrapped standard errors[43,44]. To ensure that the regression results are not due to chance, we compared the reported coefficients to those expected by chance. The results indicate the observed regressions coefficient cannot be explained by chance (Fig. 5).

Balance is significantly and positively associated with a trader's performance for both the Logit and non-parametric regression ($p < 0.001$) (Fig. 6). The relationship is robust to controls for market uncertainty, time period fixed effects, and individual trader effects (average trade value, number of active trading days, degree centrality), for each period (Fig. 6a). This result demonstrates that traders typically perform best, i.e., benefit from a "hot hand", when they have relatively high balanced relationships. In fact, balance presents a superlinear effect. This strong positive relationship holds for over 75% of the data. The change from medium to high balance is associated with an almost 30% increase in profits. For the bottom 25% of the data, a change in a trader's level of balance has no association with their trading performance (Fig. 6b). This suggests that low levels of balance are unrelated to trading success but from medium to high levels of balance, any increase in balance is positively and significantly associated with increases in performance. Our result is consistent with synergy theory[45] and the classic Morrissette et al. study[46]; however, to the best of our knowledge, this is the first time the relationship has been tested on a longitudinal dataset.

**Discussion**

Balance theory provides an explanation for why interpersonal sentiment networks shift towards states of structural balance.

**Table 3 Stationary distribution of the average Markov chain overall periods**

| Triad type | 300 | 102 | 003 | 120D | 120U | 030T | 021D | 021U | 012 | 021C | 111U | 111D | 030C | 201 | 120C | 210 |
|---|---|---|---|---|---|---|---|---|---|---|---|---|---|---|---|---|
| Stationary probability | 0.02 | 0.20 | 0.75 | 0.00 | 0.00 | 0.00 | 0.00 | 0.00 | 0.02 | 0.00 | 0.00 | 0.00 | 0.00 | 0.01 | 0.00 | 0.00 |

The stationarity of the null triad state suggests that forbidden triads remain in the network

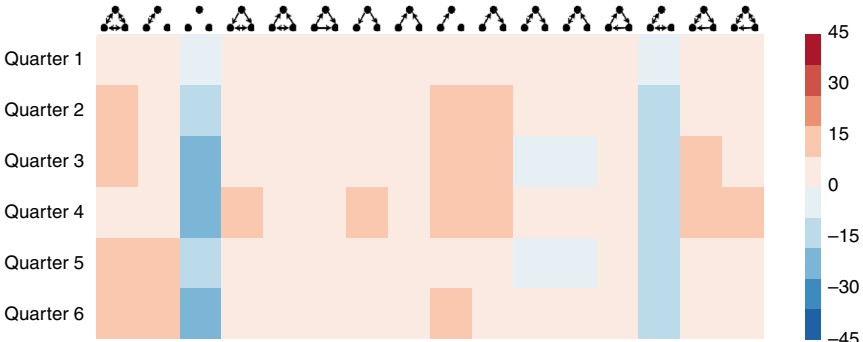

**Fig. 3** The difference in the number of standard deviations of the observed network from 10,000 suitably randomized networks. Warm colors mean more probable than random, while cold colors mean less probable. The observed networks are statistically and significantly more balanced than randomized networks

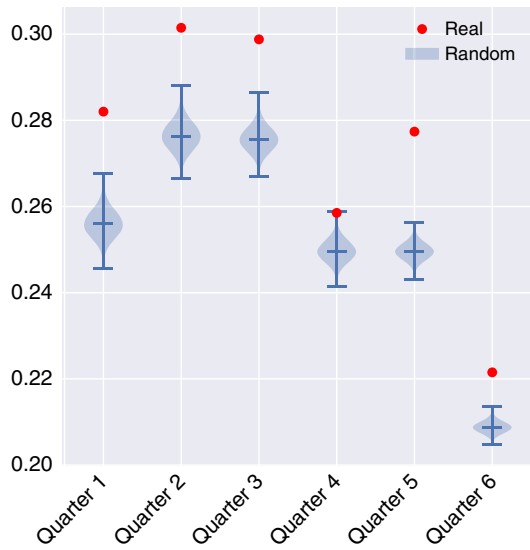

**Fig. 4** Comparison of the observed balance in the system, $b_t$ to the expected $\hat{b}_t$ (CI is shown) in each time period indicates that the observed system is in a greater state of balance than would be expected in a comparable randomized network

Little quantitative work has tested the theory's underlying premise in dynamic networks or the presumed link between balance and performance. We analyzed a social network of day traders at a hedge fund using the full corpus of instant message exchanges to infer positive and negative interpersonal attachments over a 2-year time period. Our conclusion is that sentiment networks tend toward attractor states in which violations of the SBT theory's four axioms are removed more frequently in the observed network than expected by chance. However, there are novel findings about the temporal process of balance. We find that already balanced triads tend to be highly stable. Thus, once a triad transitions to a balanced state, it tends to remain in balance due to high probability of self-transition for balanced triads[38] (see

Fig. 2). For unstable triads, different triads have different transition propensities and certain forbidden triads persist in the system, i.e., the null triad, which had been predicted by Heider[39], and introduced in a subsequent balance theory model by Davis et al.[38].

The development of structural balance theory has strictly focused on the structure and evolution of sentiment networks. This focus is motivated by a beautiful correspondence between its elementary axiom set and the macro-topology of a sentiment network. An untested premise of SBT is that it is related to performance, an implication with important consequences for the organization and economics of teams, networks, and other collectives. Research on organizations suggests that individuals choose balance at the expense of talent because individuals favor liking and trust ("lovable fool") over talent and skills ("competent jerk")[41]. By contrast, our test found that the hot hand is more likely to take place when an individual is in structural balance than out of structural balance. One explanation for the finding is that high balance and talent are not mutually exclusive. If balanced relationships result in more trustworthy information even if not with the best informed or most skillful individual, they may reduce verification costs. In our context, lower verification costs can mean trading is more responsive to market opportunities[25]. Further, balanced ties may offer more social support, reducing the emotional highs and lows that undermine risky decision-making or periods of poor trading[47,48]. In particular, both the information needs of successful trading decisions, facilitated through instant e-communication, and the emotive nature of trading relationships emphasize the need to develop balanced ties to support collaboration and communication among traders over individualism or isolation. Conversely, traders with more strained relationships may need to expend a greater proportion of their energy managing their non-cooperative relationships. In our study, we find evidence suggesting that the expulsion of energy towards managing non-cooperative relationships can detract from people's abilities to effectively utilize their balanced relationships. More broadly, beyond the context of risky decision-making, these findings suggest that future research should further investigate the

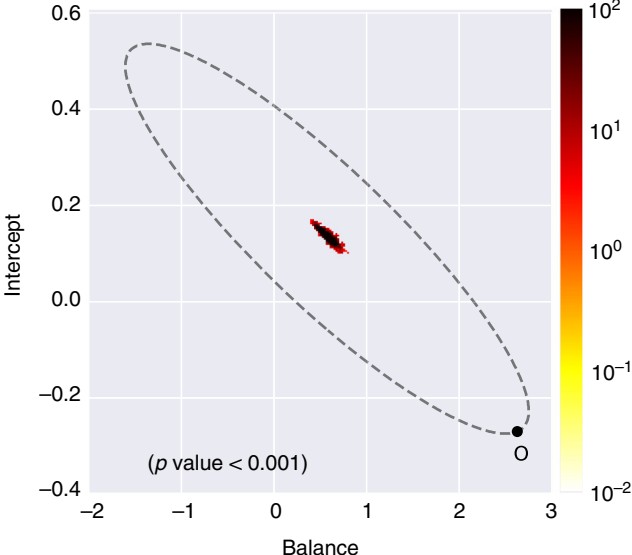

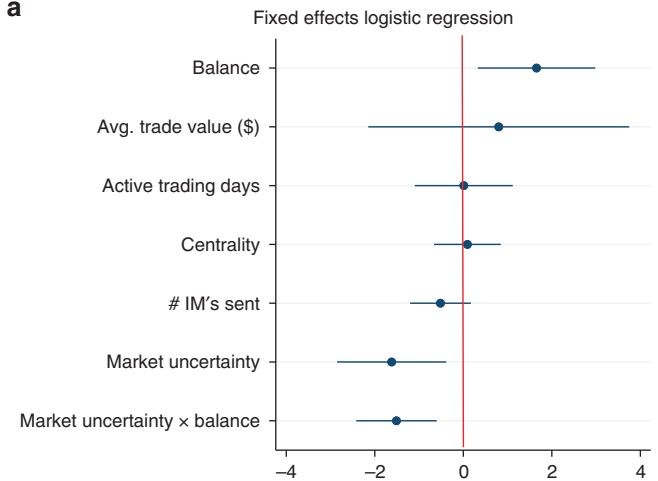

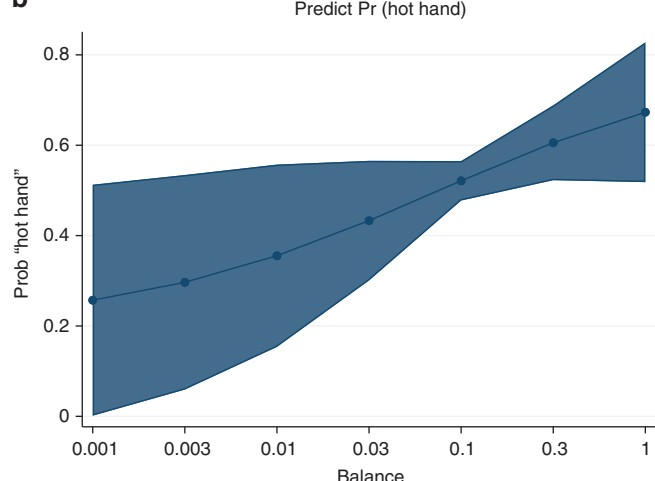

**Fig. 5** Hot hand Logit regression social balance significance, $P < 0.001$. This figure shows the results of 10,000 null models randomizing the networks, point $O$ represents coefficient from the observed networks, dots in the middle of oval represent those of randomized networks and the color shows their distribution. The coefficients for the observed model are significantly different from randomized networks with the same in and out-degree distribution. It depicts the observed balance-hot hand relationship cannot be explained by chance

mechanisms by which balanced ties might improve or hinder other performance outcomes such as creativity and innovation, negotiations, conflict resolution, and pro-social behavior. For example, balanced ties might weaken the creative tensions that promote breakthroughs in science, art, and philosophy[49].

Building on our findings, future work might begin to investigate exogenous drivers of network dynamics. SBT theory has been endogenously focused on internal group dynamics. How and whether external forces are related to balance has been left largely unaddressed despite evidence that external conditions affect how people value and interpret their relationships. Our regression analysis showed that balance was sensitive to the overall volatility in the market. Experiments could be devised to explore the mechanisms by which interpersonal attachments change over time in complex collectives that include social hierarchy, norms and rules for interaction that force the mixing of friend and enemy relationships, or where relationships are utilitarian in nature first.

## Methods

**Trade data and trader performance**. We observed all of the dynamic sentiment network of day traders at an anonymous trading firm from 1 October 1 2007 to 31 March 31 2009. Day traders keep short-term positions and do not hold inventories of stocks; they enter and exit positions each day, normally between 9:30 AM and 4:00 PM. We observed these traders trading ~4500 different stocks over various exchanges, which suggests that they sample a large part of the market. As in most trading firms, traders do not trade every day of every week for various reasons. We analyzed all of the >1 million intra-day stock trades of these day traders and their >2 million instant messages exchanged across their networks. The performance data were calculated using standard industry metrics.

**Instant messaging communication networks**. To identify IM's containing social information, we used a dictionary-based approach, comprised of terms from the NASDAQ stock exchange and IG trading glossary to differentiate between IM's containing financial and personal information. To classify information exchanges, we tagged all IM exchanges that contained at least one word from the financial dictionary. The average IM is ~6 words in length, consequently each one represents

**Fig. 6** Positive classical structural balance and having the "Hot Hand". **a** Shows coefficient estimates from an individual trader and period fixed effects for Logit regression. **b** Margins plot of the predicted relationship between the level of structural balance and having the hot hand based on the non-parametric regression. Values are means and 95% CI. Balance presents a superlinear effect. Positive relationship represents 75% of data. Traders trade best (i.e., have the hot hand) when their balance is relative high. The increase from medium to high balance has relatively high association of profits of nearly 30%. x-axis is reported as $e^{log(balance)}$

important information about the likely instrumental or social intent of the IM. A sample of 1000 IM's were selected at random to validate the classification method. In the validation method, an IM tagged as having at least one word from the dictionary were read by a research assistant who agreed or disagreed that the IM represented an financial rather than a social IM.

After extracting the content of all messages to isolate social communications from instrumental communications, we used's[35] method of estimating the strength of a social relationship from digital communication data. The method identifies positive edges between traders by comparing pairwise communication intensity levels in the observed social network vs. a statistical null-model of IM communication, where the observed pairwise level of IM exchange was randomized 10,000 times. For every period, an edge was defined as positive if the total number of IM's exchanged between two traders exceeded the random intensity scores at the $p<0.01$ level of significance. Following prior research, edges between traders that are below the threshold are defined as non-positive or negative ties[35,50].

Albeit balance theory research has defined non-positive edges as negative, we conducted a robustness test within our setting. To check the validity of our measurement to misclassifying ties as negative when they should be positive, we purposefully converted multiple (10,000 replicas) 5% samples at random in the observed data from negative to positive edges. The reported results were robust to these measurement tests suggesting that the definition of an edge's polarity is

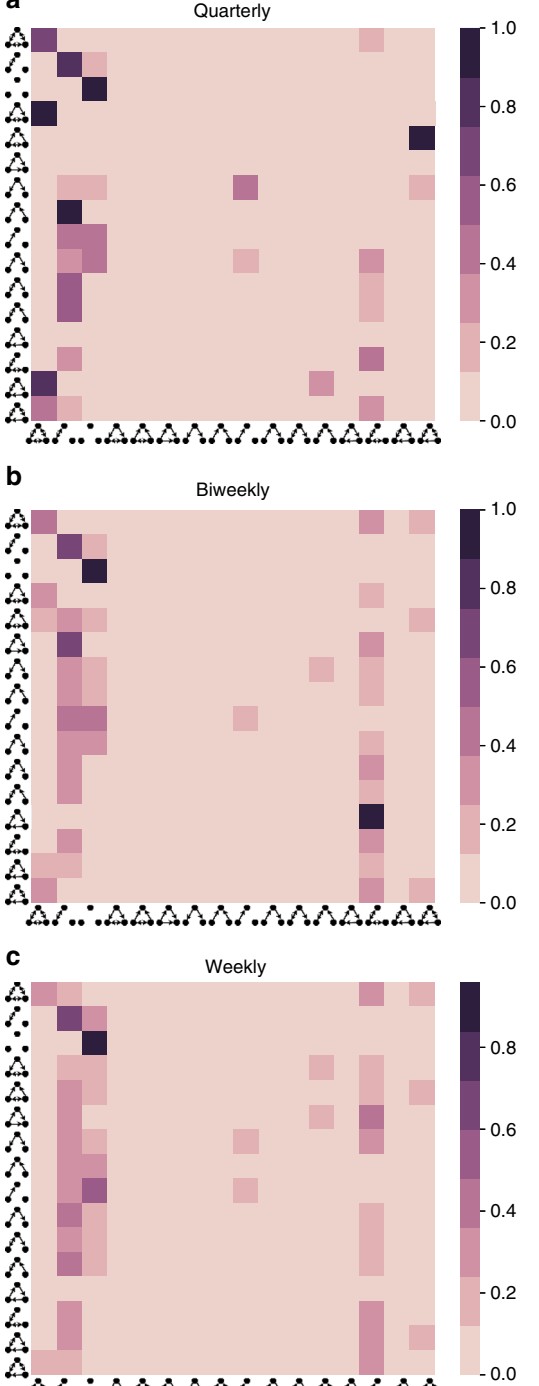

**Fig. 7** The aggregated stochastic Markov transition matrix of all periods together, with **a** quarterly, **b** biweekly, and **c** weekly periods (i.e., the quarterly matrix shows the average of all matrices given in Fig. 2). Other designations are as Fig. 2). The transition probabilities are robust regardless of the choice of the period interval

robust to significant measurement. Changing the polarity of edges at random in the preceding way up to 20% did not change the statistical significance or pattern of reported results.

The data setting meets the requirements of mutually acquainted individuals, for which traders develop positive and negative sentiments towards each other, not neutral attitudes[10,11,13,15]. This assumption is consistent with the cognitive science literature on the automaticity of attitudes[51–54] and instantaneous formation of impressions[55], as well as the communication literature examining ease of relationship formation over electronic communication[56,57], for which use of computer screens is essential to day traders' work activities. Prior research

examining negative ties as avoidance behaviors has also measured the absence of an edge as a negative tie[58,59]; we use this approach to be consistent with the prior work.

In addition to the volume method, we used a simple threshold cutoff to define an interpersonal relationship, where the presence of an edge corresponded to a trader sending at least 1, 5, or 10 messages to another trader, respectively. Our findings are robust to methods and thresholds (Fig. 1a).

**Measure of classic structural balance.** To quantify structural balance of the firm over time, we divided the entire observation period into six quarterly intervals, $t$, and defined a measure to capture the degree of balance at each quarter. For each period, we computed the ratio of balanced triads to the total number of possible triads with the measure, $b_t$. We used the "classic" model of structural balance, for which balanced triads were defined as the count of 300 and 102 triad types because both configurations satisfy all of Balance Theory's four rules (Table 1). To verify our selection of quarterly time intervals, we also analyzed the data using monthly, bi-monthly, biweekly, and weekly time intervals and the results were robust to period interval (Fig. 7). The distribution of triads, transition matrices, and stationary distributions were similar in these results except that in biweekly and weekly periods, there were more 003 triads, and the probability transition to the 003 triad is higher, which is expected given the the smaller time interval (i.e., 5 or 10 business days), during which traders can IM each other. On average, traders exchange messages with two to three other contacts each week.

To ensure that our observed triadic network configurations could not be explained by chance, we constructed null models to compare the observed likelihoods of the balanced triads in the network, $\hat{b}_t$, to randomized networks, $b_t$, using[60], with dyadic and triadic configurations.

**State transition probabilities.** For each consecutive observation period, $(t, t + 1)$, we compute $T_{ij}(t)$, which is the number of triads of type $i$ that moved to type $j$ from period $t$ to $t + 1$. Thus, row $i$ sums to $T_{i\cdot}(t)$, which is the number of triads of type $i$ at time $t$, while $T_{\cdot j}(t + 1)$ is the number of triads that have transitioned to type $j$ at time $t + 1$. Using $T_{ij}(t + 1)$, the transition probabilities, $p_{ij}(t)$ can be estimated to obtain the transition probability matrix. These quantities can be arranged in a matrix and normalized by the sum of every row. Therefore, we have row-stochastic transition matrix $P$ where each $p_{ij}(t)$ is conditional on $i$ only, and not on prior states occupied by the triad. By the Markov property, they are identical for all triads, and they converge to a stationary distribution. The stationary distribution of a Markov chain is the probability distribution that a system remains unchanged as time progresses. Mathematically, it is computed as the normalized left eigenvector corresponding to the eigenvalue of 1 of the row-stochastic transition matrix[61,62]. We compute it for every transition between two subsequent periods (Fig. 1b, c). The stability ratio examines the likelihood for every transition in the observed transition matrix to happen by chance. It compares every element of the observed matrix ($16 \times 16$ elements) to the corresponding element of 10,000 transition matrices computed on randomized networks[60] to determine the ratio of transitions in the observed matrix that are statistically significant for each transition period (Table 2).

Furthermore, we derive a triad count ratio, $c_{xt}$, for each triad configuration, $x$, in each period, $t$, to examine the distance between the current state and the stationary probability distribution for each triadic configuration and each period. Specifically, for each of the 16 triadic configurations, $x$, the triad count ratio is computed as the number of triads with configuration $x$ over the total number of triads for the given period. For each transition period, we compute the triad count ratio, $c_{xt}$ for each of the 16 triad types and compare it to the corresponding triad count ratio in the stationary probability distribution. A high degree of similarity between the two ratios indicates that ties are being reconfigured in a consistent manner that moves the system towards the stationary probability distribution.

Although the sentiment networks are fundamentally dynamic, our state transition analysis is insensitive to traders' entrances and exits. To extract communication networks in each period, we only take into consideration those traders who have traded in the respected time. Then for every two subsequent period, we compute transitions of triads for traders who exist in both communication networks.

**Network triads comparing observed to randomized networks.** We compare the structural patterns of interconnections in our observed networks to randomized networks[60]. For a stringent comparison, we use randomized networks that had the same single-node in- and out-degree characteristics as the corresponding node in the real network, as well as the same dyadic subgraphs as the real network[60]. This is attained through repeatedly swapping randomly chosen pairs of connections ($S1 \leftrightarrow T1$, $S2 \leftrightarrow T2$ is replaced by $S1 \leftrightarrow T2$, $S2 \leftrightarrow T1$). Swapping is prohibited if either of the connections $S1 \leftrightarrow T2$ or $S2 \leftrightarrow T1$ already exist or these edges share nodes. The same procedure is applied for mutually connected pairs of nodes. Unlike the Milo et al.'s work[60], in this study[10], the network is fully connected, and we focus specifically on 16 directed and signed triad configurations with exactly three nodes. Also networks are not static but dynamic and we focus on the transition of triads over time. Results show that the triad probabilities in the randomized network are significantly different than the observed network ($p < 0.01$).

**Structural balance and performance**. We define a trader with a hot hand as a trader that made better than average profits over the quarterly observation period where high and low profit was split at the mean profit. To examine the robustness of the association between balance and individual relative performance (i.e., hot hand) to other potential influences, we perform the same analysis with controls, as stated in the text.

**Reporting summary**. Further information on research design is available in the Nature Research Reporting Summary linked to this article.

## Data availability
Data are available from the trading firm, which retains ownership over the data. The company should be contacted directly for accessing the data for further research purposes.

## Code availability
The source code is publicly available under link https://github.com/omid55/longitudinal_structural_balance_theory.

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

## Acknowledgements

This work is supported by US Army Research Laboratory and US Army Research Office (grant# W911NF-15-1-0577), by the National Science Foundation (grant# DGE-1258507), and by UC Multicampus-National Lab Collaborative Research and Training (grant# LFR-18-547591).

## Author contributions

O.A. and J.N.L. analyzed the data and performed the experiments. N.E.F. conceived the original idea. B.U. led the study. F.B. and A.K.S. provided critical feedback and helped shape the research. All authors substantially contributed to the design of the analysis and the writing of the manuscript.

## Additional information

**Competing interests:** The authors declare no competing interests.

