## [Peer Review File · Nature Communications]

Reviewers' comments:

Reviewer #1 (Remarks to the Author):

It is with great interest that I have read this manuscript on the relations between structural balance in performance in decision making.

The paper focuses on an important topic and considers an original, high-quality dataset to tackle a long-standing problem in the social sciences.

If, at the first sight, the paper appeared well-written and compelling, I have been facing increasing difficulties and uncertainties when digging into the methodology. If I am not univocally against the publication of this paper, my opinion is that it should undergo substantial rewriting to satisfy the publication criteria of NatComm.

Besides some misprints here and there, the paper is overall well-written. However, my understanding is that the claims and motivations of the paper do not fit with data at hand nor their analysis. The whole paper builds on the notion of structural balance, defined in situations when networks have positive and negative interactions, and stating that certain types of configurations are more stable than others. From my understanding, the data analysed by the authors only allows to infer the existence, or not, of a connection; and I do not agree, without additional arguments/evidence, that the absence of a link signifies that it is negative. As it is, the paper is thus about the stability, or not, of standard motifs. Nothing more. In a system of 60 nodes, it is expected that a large number of missing edges is simply due to sparsity, or different interests, and not enmity. In a real-life sparse system, the absence of connections is the norm more than the exception.

Additional confusion comes from the fact that directed motifs are considered while, if I am correct, standard balance theory deals with undirected networks. Did the authors check that some of their patterns did not emerge from simple mechanisms like reciprocity? Or social status, as in the works of Leskovec on signed networks?

The fate of 003 configurations is also very intriguing. While the Markov chain indicates their stability, Fig 3 shows that they are under-represented. In any case, this is intriguing as 003 configurations are expected to be unstable in balance theory. Moreover standard balance theory predicts that the system either converges to global consensus, or to two groups/factions, but neither of those states has 003 configurations.

As a minor note, it would be helpful that the authors provide more basic statistics of the networks in each time window.

Also, I am unsure of the message of Figure 4. Does balance increase or decrease?

Finally, I was also surprised that certain configurations have no entries in the Markov chain matrix (empty lines). Is there a reason for this?

Reviewer #2 (Remarks to the Author):

Summary:

This paper quantifies the tendency towards structural balance in the communication networks of financial traders. Specifically, the authors create communication networks from the instant messages of day traders and explore the temporal transitions of the motif structure around each trader. Remarkably, there was a significant relationship between the structural balance of a trader's communication network and sequences of "hot hand" performance in the market.

Overall the paper offers a novel analysis and adapts several interesting techniques from computational social science to explore a topic of interest to an interdisciplinary audience. However, its presentation could be improved, as I discuss below. I would be delighted to look at the revised version, in due course.

Major notes:

1) I am having difficulty following the network construction process, I wonder if the authors could improve on that. To be specific, here are some guiding questions:

- ~ "After extracting the social messages" - does this mean only social messages were used? Why?
- ~ The volume method from ref 27 produces an undirected graph and does not make a reference to a random null model, yet the authors work with a directed graph. Please elaborate.
- ~ The null model is described as "the observed pairwise IM exchange was randomized 10,000 times". Does this mean the sender and receiver were randomized? From what I understand, the scenario requires a weighted-degree conserving randomization in which the frequency of in-coming and out-going IMs is conserved for each individual.
- ~ The authors use the name "sentiment network", but it is more appropriately an IM-frequency network (sentiment is typically used in reference to the content of the messages).
- ~ Is the number of IMs sent consistent between the six quarters?
- ~ How much of the motif structure can be attributed to the bursty nature of human communication?
- ~ What is the distribution of motifs observed at each quarter?

2) I like how the authors employed a Markov transition matrix between network motifs to capture changes in structural balance. However, it is a little difficult to understand the exact analysis conducted at this stage and why it is identifying significant features of the communication network.

Specifically:

- ~ The matrices are very sparse, is this largely because of the underlying distribution of motifs? How far are these matrices from transitions between independent distributions of motifs at each quarter?
- ~ There are several unexpected transitions present (ex from 300 to 201, 210 or from 102 to 003). Can the authors shed some light on why they are observed?
- ~ can you put Table 2 into a context? The transition 2-3 seems very large for such a sparse matrix. Furthermore, I noticed that the authors only visualized the 3 transition matrices with smallest normed distance. What is the structure of the other two transitions (2-3, 4-5)?
- ~ Figure 2 is the only visual evidence used to support the claim that structurally balanced motifs 300, 102 are stable (as well as the null 003). Can this statement be quantified. How much more stable are these configurations than expected by a random time series (say when the time-stamps of the IMs are randomly shuffled)?
- ~ was the stationary distribution referenced in Table 2 the same for every row (ie the stationary distribution of the 5-step markov chain) shown in table 3, or was a stationary distribution found for each of the 5 transition matrices? Can you walk through the meaning of the distribution given in table 3?

Specific notes:

- ~ Fig 3. Suitably randomized networks are not clear. What procedure?
- ~ Fig 4. Please place error bars on the random network points. (Or is that what the lines denote?). Connecting the 6 quarters with a line is misleading, perhaps consider a bar-chart to indicate all quarters were randomized independently.
- ~ Did the regression analysis control for IM frequency? If the number of IMs is related to

performance, and IMs tend to obey structural balance, then one might expect to see a relationship between structural balance and performance.

~ Fig 6 C (note, there is no b) is very awkward. Only 3 points are shown, and the x-scale is not clear. The y-labels do not clearly show the decimals.

~ There are several grammar mistakes and typos, including pg 6, para 4,

~ avoid double negative: not dissimilar, page 6, para 5

Reviewer #3 (Remarks to the Author):

The authors investigate how affective relationships among 66 stock traders affect their trading performance, over a period of two years. The interrelation among brokers is built by using the instant messages they exchange daily to gain information relevant to their trading. The traders' performance is monitored from stock market data and further used to investigate whether balanced relationships make traders to perform better or worse.

The study analyzes the largest longitudinal data assembled so far, as reported by the authors, and this favors a unique analysis of the sentiment network across a large time frame. Structural balance theory (SBT) is used as modelling framework for the process under study. The authors claim to find a tendency for the sentiment network to transition into states of greater balance over time. Furthermore, they conclude that the trader's degree of structural balance is positively correlated with the trader's performance.

The findings of the study are interesting to understand how social relationships can affect performances. However, some parts of the paper deserve further clarification and limitations of the study should be better addressed. In some part, results are overstated or not fully discussed in their limits and it is hard to draw a final conclusion on the validity of the study. I further elaborate below on these salient points.

a) The authors claim that the transition probabilities are relatively stable across transition periods and they use, as support of this statement, the L2 norm distance between the current state and the stationary probability distribution (SPD), see Table 2. Within the whole text, though, there is no mention of how the SPD is obtained, which is a crucial quantity for all the results which come next and, therefore, their interpretation is more difficult.

b) Structural balance theory predicts that only the triads 300 and 102 should exist in a sentiment network with no violations of the axioms, as remarked by the authors. From the analysis of the stationary states, the authors conclude, indeed, that the triads 300 and 102 have higher occurrences of balanced states. This is partially true from my examination of the results. Indeed, from Table 3 one reads that the state with the highest SPD is the null state, with $SPD = 0.68$. According to structural balance theory this state is unbalanced and, therefore, should have a null or very small SPD, from my understanding. The authors comment this fact by stating that the high SPD of this state suggests that the network remains sparse over time, which is true, but to me this also signals a limitation of the SBT in the prediction of the stationary states. Furthermore, from Table 3, one reads that the states 012 and 201 (predicted unbalanced by the SBT) have SPT of 0.02 and 0.04, respectively, which are twice and four times higher than the SPD of the state 300, with $SPD = 0.01$, which is predicted balanced by SBT. Again, this signal a failure or at least a limitation of SBT in the prediction of the stationary states

that it is not only worth mention but, perhaps, deserves some further analysis and explanation.

c) To test whether their results are due to chance, the authors compute the likelihood of each triad from suitably randomized networks, see Fig 3. They find that the actual network has a lower occurrence of null triads and, thus, the high SPD of the state 003, cannot even be explained by random effects in the data. How do the authors interpret this result together with the remarks raised in point b) above about the SPD of state 003?

d) To build the sentiment network and compute the transition probabilities the authors fix a quarterly period (four months) as time-window to build the sentiment network and compute the transition probabilities. The construction of the network strongly depends on the time-intervals employed to partition the whole observation period. In Methods, the authors point out that the analysis of data using monthly and bi-monthly time intervals produced similar results. As the authors describe in the text, the traders open and close several trading position every day and they have high volume of instant messages that exchange every day. For this reason, and due to the nature of the problem, certain processes might happen at a much shorter time scale than a month. I wonder if the authors have tried to analyze the data also with weekly and bi-weekly time intervals and, if so, if they have found differences in the stability of the states and in the conclusion of the study. I believe the effect of the time-interval chose to analyze the data on the result to be a very important issue which should be discussed extensively.

e) Fig. 4 shows an overall decrease of the classically balanced ratio of the real network. The authors comment that this decline corresponds to the uncertainty due to the 2008-09 financial crash. The figure shows, though, that the same decline is also a feature of the balance ratio of randomly generated networks. These networks, being randomly created, should not be sensitive to the actual data and, thus, to the financial crash of 2008-09. So, I wonder, how are the random networks generated in this study? Why such decline does effect also these random networks?

Statement of Revision

Structural Balance Emerges and Explains Performance in Risky Decision-Making

We would like to thank the reviewers for reviewing our paper. We appreciate the thoughtful and constructive comments from all reviewers. In the following, we provide a detailed account of all the changes that we have made in the revised version of the paper. We have structured this list in separate blocks, corresponding to the comments made by the referees. The referee's text in is blue followed by our response is in black. Also, in the revised manuscript all the changes resulting from the revision are in red followed by their page and line number in the main manuscript. We would like to thank all the reviewers for their thoughtful and constructive remarks.

Comments by Referee # 1

It is with great interest that I have read this manuscript on the relations between structural balance in performance in decision making. The paper focuses on an important topic and considers an original, high-quality dataset to tackle a long-standing problem in the social sciences. If, at the first sight, the paper appeared well-written and compelling, I have been facing increasing difficulties and uncertainties when digging into the methodology. If I am not univocally against the publication of this paper, my opinion is that it should undergo substantial rewriting to satisfy the publication criteria of NatComm. Besides some misprints here and there, the paper is overall well-written. However, my understanding is that the claims and motivations of the paper do not fit with data at hand nor their analysis.

Thank you for your review. We hope our responses to your specific concerns will clarify the claims/methodology of the paper. We believe, the changes that have been made in response to your comments, have served to strengthen the paper.

[R1: 1] “The whole paper builds on the notion of structural balance, defined in situations when networks have positive and negative interactions, and stating that certain types of configurations are more stable than others. From my understanding, the data analysed by the authors only allows to infer the existence, or not, of a connection; and I do not agree, without additional arguments/evidence, that the absence of a link signifies that it is negative. ‘...’ In a system of 60 nodes, it is expected that a large number of missing edges is simply due to sparsity, or different interests, and not enmity. In a real-life sparse system, the absence of connections is the norm more than the exception.”

We want to thank the reviewer for pointing out a very fundamental comment. This was a very important to us as well and we gave it a lot of thought.

Below, we describe three lines of SBT research that define edges in the same way we have defined edges in our paper. The lines of reasoning are a) theoretical b) setting c) and empirical validity check:

- (i) Theoretical: The approach taken in our work is consistent with previous work that has defined negative ties as the absence of an edge. Original SBT research (Davis, 1967) defined strong ties as “positive” and weak or absent ties as “negative” [1]. More recently an active line of research studying networks in business organizations has continued and further substantiated the theoretical basis for defining ties in the manner we have defined in the paper (i.e., as negative attitudes, such as avoidance, distrust and dislike) [2]. Labianca et al. (1998) writes “the prefer-to-avoid response was considered indicative of a negative relationship at the interpersonal level” [3].
- (ii) Setting: As you mentioned, in large systems the absence of an edge may be simply due to sparsity because it is not possible for all individuals to become mutually acquainted; similarly, in hierarchical structures, some ties are mandated, such as those between supervisors and subordinates, whereas other ties do not have the opportunity to develop, such as those between workers in different business units or geographies; also, in

settings where individuals perform independent work, there is seldom a need for collaboration between the members of a system. However, these systems do not describe our setting.

The day trading firm we study has ~ 60 individuals, is non-hierarchical, and exists in a high-stakes, competitive environment where communication and social relations are necessary and can contribute directly to performance. This data context has three implications on the formation and persistence of positive and negative ties.

- The small size of the firm indicates that every individual is mutually acquainted with each other. By following the firm over 1.5 years, we have confidence that each individual has the opportunity to become acquainted with one another. Thus, the absence of tie suggests intentional avoidance.
- The non-hierarchical structure of the trading firm, suggests that traders are free to turn to their peers to seek and provide advice.
- The nature of the work (i.e., highly uncertain, high risk/high reward, competitive, high stress) make it highly expected that traders develop interpersonal relationships for emotional support. This aspect, combined with the fact that screen time is crucial in these traders' work, indicates that most communication between traders occurs electronically over IM. The ease of communicating over IM indicates that any trader can talk to and develop a relationship with anyone else in the firm. Additionally, research has shown that the frequency of IM communications is (nearly) equivalent to the frequency of face-to-face interaction between two individuals [4], providing support for the use of electronic communication to study human interpersonal sentiment networks. Together, these factors provide support for the inference that the absence of an edge is indicative of a negative tie.

(iii) Empirical Validity Check: We conducted a validity check of our statistical method for measuring positive vs. negative ties. In this test, we intentionally introduce error into the observed network, by converting 5% of the negative ties to positive ones at random and vice versa. We compute the transition matrix between consecutive time periods for the perturbed networks (i.e., noise-added networks) and repeat this process 10,000 times for each of the five transition matrices. Then, for every period, we compare the perturbed transition matrix to the 10,000 randomized transition matrices computed from the perturbed networks. Next, we apply a statistical test and compare every element of the observed perturbed matrix (16×16 elements) to the randomized transition matrices to determine the ratio of transitions in the observed matrix that are statistically significant after error is introduced into the network ($p < 0.05$). The reported results hold up until about 20% noise. This high percentage indicates that the results are robust to noise (error) in the attribution of positive/negative ties.

The arguments in (i)-(iii) indicate that in contrast to other networks (e.g., large online communities, online social networks), the non-positive edges are likely associated with negativity, rather than simply sparsity or different interests. Also importantly, note that balance theory does not strictly focus on enmity. Negative relations may be weak or strong.

We have strengthened the manuscript by adding citations supporting on the argument of negative ties in page 6, line 88 and by reporting the results of the validity check in page 6, line 77.

[R1: 2] “As it is, the paper is thus about the stability, or not, of standard motifs. Nothing more.”

In scientific literature, motifs have a specific conceptual association with them as the building blocks of complex networks and refer to n -node subgraphs (Milo et al. 2002) [5]. We are not using motifs in that manner but use motifs to describe a set of triads that are particular to balance theory and to examine how these sixteen triads (Fig. 1 in the manuscript) evolve over time in a social system as a test of balance theory's propositions. For example we are interested in examining whether certain triads resolve into balanced states, which is not part of the study of motifs.

We have referred to these triads as motifs because it is done so in social science literature but we now realize that this terminology is confusing in the science & engineering literature. Consequently, we are now removing the word

motifs from the paper and are just focusing on triads, and their balanced vs. unbalanced states.

We also clarified how balance theory and the study of triads differs from study of motifs in the revised manuscript, in page 7, line 62.

[R1: 3] “Additional confusion comes from the fact that directed motifs are considered while, if I am correct, standard balance theory deals with undirected networks.”

Respectfully, Heider (1946)’s classical balance theory is defined on directed networks [6]. Subsequent work by Cartwright and Harary [7] (see Fig. 2 and Fig. 6 in their paper) and Johnson (1985) [8] are also based on directed networks that are consistent with the triad definitions used in Fig. 1 of the original manuscript.

The confusion might originate from the fact that balanced networks are symmetric, but during the network evolution to balance, the networks are not necessarily symmetric. Symmetric networks are sometimes treated as undirected; however, undirected networks (i.e. symmetric networks) are the endpoints of the evolution to balance.

Per this comment, we have now clarified in the revised manuscript that structural balance theory is defined on directed networks, in page 1, line 21.

[R1: 4] “Did the authors check that some of their patterns did not emerge from simple mechanisms like reciprocity?”

Thank you for your comment. Reciprocity is fundamental to most human relationships. Based on literature, we know by definition the evolution to a balanced structure must depend on reciprocity but not all reciprocal (symmetric) networks are structurally balanced. For instance, triad 201 is reciprocal but not balanced. The information on transition toward reciprocity is given in the transition matrix in Fig. 2 of the original manuscript and also Fig. (4) in this document. In most periods, non-balanced triads, including the mentioned triad, have small self transition probabilities and they are more likely to transit to one of the three balanced triads based on Davis et al. [1].

[R1: 5] “Or social status, as in the works of Leskovec on signed networks?”

Thank you for suggesting Leskovec’s work [9] on signed networks that compares the opposing theories of balance and status. That said, the trading firm selected for our empirical setting is a very flat, non-hierarchical environment. The day traders have the same formal status/rank, and have autonomy over the stocks they trade as well as their own trading profits and losses. Given this setting, we think that balance theory is a more compelling framework. Also, electronic communication, such as IM’s tends to minimize social status differences between individuals (e.g., equalization phenomenon [10]).

Per this comment, we have clarified our empirical settings in the manuscript, in page 2, line 22.

[R1: 6] “The fate of 003 configurations is also very intriguing. While the Markov chain indicates their stability, Fig. 3 shows that they are under-represented. In any case, this is intriguing as 003 configurations are expected to be unstable in balance theory. Moreover standard balance theory predicts that the system either converges to global consensus, or to two groups/factions, but neither of those states has 003 configurations.”

We want to thank the reviewer for pointing out an important finding that could be further clarified.

It is correct that classical balance theory does not allow for triad 003. However, as been pointed out, in this study, the results indicate that triad 003 is prevalent in the stationary distribution. As we mentioned in the original manuscript, the explanation of this finding comes from the seminal work by Davis [1]. Davis (1967) relaxes the classical balanced theory by neglecting the fourth rule: “enemy of enemy is friend”. With that that it allows for triad 003; hence, we can have multiple groups/factions with all positive ties within the group and all negative between group ties. Heider himself writes ‘if two negative relations are given, balance can be obtained either when the triad relationship is positive or when it is negative, though there appears to be a preference for the positive alternative.’ [11]. Heider

foresaw this phenomenon originally [11], Davis was the first to introduce that as "clustering model" [1], but their overlooked prediction turns out to have empirical support for the first time in a longitudinal field setting.

Per this comment, we have now strengthened the manuscript by adding the theoretical backgrounds on the existence of triad 003 in our findings in page 3, line 25.

[R1: 7] "As a minor note, it would be helpful that the authors provide more basic statistics of the networks in each time window."

We have added the pertinent statistics to the manuscript for the mean (standard deviation) of number of trader IM's, outdegree, clustering coefficient, and closeness in the network over the 6 periods (in page 2, line 35).

[R1: 8] "Also, I am unsure of the message of Figure 4. Does balance increase or decrease?"

Fig. 4 shows the degree of balance in the observed network compared to a randomized network with the same number of nodes and edges. The declining ratio indicates that the number of edges in both the observed and randomized networks are declining in each period. That said, we show that the observed network still has a greater degree of balance (i.e., more classically balanced 301 and 102 triad configurations) than the randomized networks.

Per the comment, we have also clarified this point in the revised manuscript, in page 4, line 18.

[R1: 9] "Finally, I was also surprised that certain configurations have no entries in the Markov chain matrix (empty lines). Is there a reason for this?"

Each of the possible 16 configurations, defined based on classical balance theory (see Table 1 in the manuscript) has an entry in the Markov chain matrix. However, many of these configurations occur with a very low (near 0) probability. This is consistent with balance theory's assumptions that non-balanced states occur with very low probability.

We have also clarified this point in the revised manuscript, in page 3, line 17.

Comments by Referee # 2

This paper quantifies the tendency towards structural balance in the communication networks of financial traders. Specifically, the authors create communication networks from the instant messages of day traders and explore the temporal transitions of the motif structure around each trader. Remarkably, there was a significant relationship between the structural balance of a trader's communication network and sequences of "hot hand" performance in the market.

Overall the paper offers a novel analysis and adapts several interesting techniques from computational social science to explore a topic of interest to an interdisciplinary audience. However, its presentation could be improved, as I discuss below. I would be delighted to look at the revised version, in due course.

Thank you for your review. We hope our responses to your comments will clarify and improve the presentation of the paper. We posit the changes that have been made in response to your comments, have served to strengthen the paper.

[R2: 1] "I am having difficulty following the network construction process, I wonder if the authors could improve on that. To be specific, here are some guiding questions:"

(i) "After extracting the social messages" - does this mean only social messages were used? Why?"

Yes, we used social messages because the basis of structural balance theory is social relationships. Accordingly, we use a dictionary-based method to classify the messages into financial and social messages. The social messages are taken into account in this study because they are the indicators of individuals' interpersonal (affective) relationships. If two traders send financial-related messages to each other, these are instrumental

relationships and are not necessarily indicative of friendship. When traders share personal/social information with each other, it indicates that there is a level of trust and friendship underlying their communication.

Per the comment, we have improved the manuscript by clarifying this point in the revised manuscript in page 6, line 72.

- (ii) “The volume method from ref 27 produces an undirected graph and does not make a reference to a random null model, yet the authors work with a directed graph. Please elaborate.”

The aforementioned method is extended to directed graphs. This method computes the zscore metric for every directed edge in the graph using the average number of social messages between every two traders. We only consider the directed edges that are statistically significant at the $p < 0.01$ level of significance. In addition, directed ties are supported by SBT, with Heider (1946)’s classical balance theory being defined on directed triads [6, 7, 8, 12]. Please also refer to the response to the point [R2: 3] raised by reviewer 2.

- (iii) “The null model is described as “the observed pairwise IM exchange was randomized 10,000 times”. Does this mean the sender and receiver were randomized? From what I understand, the scenario requires a weighted-degree conserving randomization in which the frequency of in-coming and out-going IM’s is conserved for each individual.”

”[the scenario requires a weighted-degree conserving randomization in which the frequency of in-coming and out-going IM’s is conserved for each individual...]”: This is completely correct. The network randomization algorithm preserves every node’s number of incoming and outgoing number of edges. This is attained using the swapping method described in the Milo et al. (2002) [5]. In every observed network, we repeatedly swap randomly chosen pairs of connections ($S1 \longleftrightarrow T1, S2 \longleftrightarrow T2$ is replaced by $S1 \longleftrightarrow T2, S2 \longleftrightarrow T1$) until the network is well randomized. Swapping is prohibited if either of the connections $S1 \longleftrightarrow T2$ or $S2 \longleftrightarrow T1$ already exist or these edges share nodes. The same procedure is applied for mutually connected pairs of nodes. Fig. 1 shows the condition for swapping in mutually and non-mutually edges. Consequently, we preserve both the in- and out-degree, and the number of mutual ties for each node in the network. Based on the size of our networks, we swap $\sim 49K$ pairs of edges on average. This point is also stated in the Methods section of the manuscript.

Per this comment, we improved the manuscript by adding more details about randomization method, in page 7, line 56.

- (iv) “The authors use the name “sentiment network”, but it is more appropriately an IM-frequency network (sentiment is typically used in reference to the content of the messages).”

Prior literature has defined interpersonal sentiment networks as “sentiment networks” [13, 14]. In this study, the sentiment network is used to describe the polarity of the edges of a social network, not the content of the messages. The content of the messages are used only to parse communications that are instrumental vs social in nature. Only social communications are used in the creation of the sentiment network as described in detail in the methods section.

- (v) “Is the number of IM’s sent consistent between the six quarters?”

The number of IM’s sent per individual is relatively consistent over each period as reported in Table 1.

- (vi) “How much of the motif structure can be attributed to the bursty nature of human communication?”

Although burstiness is an important aspect of human communication, we aim to reduce its impact by examining the communication networks over longer periods of time (e.g., monthly, bi-monthly, and quarterly) as well as different time periods. Per reviewer 3’s suggestion ([R3: 4]), we have also analyzed the data using bi-weekly and weekly intervals as showed in Fig. 4. The findings are mentioned in the manuscript in page 6, line 116.

- (vii) “What is the distribution of motifs observed at each quarter?”

Figure 1: Pair edge swapping condition. Swapping happens if 4 nodes are different and there is no edge on the red region. This preserves number of nodes, edges, and every nodes' in- and out-degree.

Table 1: **Number of messages in 6 quarters.**

Periods	Average #IMs per trader	Average outdegree per trader
Quarter 1	294.26	6.65
Quarter 2	209.76	6.16
Quarter 3	187.11	6.09
Quarter 4	193.72	5.79
Quarter 5	211.09	6.00
Quarter 6	271.00	5.19

The empirical distribution of triads, as expected, is a very close match to the stationary distribution already reported in Table 3 in the manuscript. Triads 300, 102 and 003 are consistently prevalent over time. As we mentioned in the manuscript, they are predicted to occur frequently by Heider and Davis et al. [11, 1].

Per the comment, we further clarified this finding in the manuscript, in page 3, line 15.

[R2: 2] “I like how the authors employed a Markov transition matrix between network motifs to capture changes in structural balance. However, it is a little difficult to understand the exact analysis conducted at this stage and why it is identifying significant features of the communication network. Specifically:”

- (i) “The matrices are very sparse, is this largely because of the underlying distribution of motifs? How far are these matrices from transitions between independent distributions of motifs at each quarter?”

In Fig. 3 in the manuscript, we compare the observed matrices to the randomized distributions of each motif (i.e., the 16 triad configurations) in each quarter, as in Milo et al. (2002) [5]. Please also refer to the response to the point [R1: 9] raised by reviewer 1.

- (ii) “There are several unexpected transitions present (ex from 300 to 201, 210 or from 102 to 003). Can the authors shed some light on why they are observed?”

Figure 2: Transition matrices for all 6 subsequent quarters.

Looking closely at those transitions, these unbalanced transitions occur with very low (near 0) probability. Although there might be few unexpected transitions, triads do not stay in unbalanced states for very long and eventually transition to one of the classically balanced triads or the null triad (allowed by the cluster model). Also, examining Fig. 2 in the dataset, we see that the probability of self-transitions in the balanced state 300 and 102, is much higher than the transitions to non-balanced states, thereby confirming our general finding that balanced states are stable.

Per the comment, we clarified this finding in the manuscript, in page 5, line 24. Thank you.

- (iii) “can you put Table 2 into a context? The transition 2-3 seems very large for such a sparse matrix. Furthermore, I noticed that the authors only visualized the 3 transition matrices with smallest normed distance. What is the structure of the other two transitions (2-3, 4-5)?”

Table 2 shows the stability of the system, as it includes the L2-norm distance of the stationary probability distribution for every subsequent period from their average. This stability suggests that traders reconfigure attachments in a consistent manner over time such that the overall system maintains a stationary probability distribution.

We were also concerned about this issue, and agree that the L2-norm distance for transition 2-3 seems large. In Table 2 of the manuscript, we now include the Stability Ratio (3rd column). The stability ratio denotes the proportion of transitions in the observed transition matrix that were statistically significant compared to the randomized transition matrices for each period. The stability ratios for each transition period indicate that traders reconfigure attachments in a consistent manner over time such that the overall system maintains a stationary probability distribution and that the observed transitions are not likely to be explained by chance. The results indicate that there are no significant differences between the Stability Ratio for transition period 2-3.

Fig. 2 now shows all five transition matrices (also in the manuscript).

- (iv) “Figure 2 is the only visual evidence used to support the claim that structurally balanced motifs 300, 102 are stable (as well as the null 003). Can this statement be quantified. How much more stable are these configurations than expected by a random time series (say when the time-stamps of the IM’s are randomly shuffled)?”

Due to the comment, we performed two supplementary analyses. First, we created a randomization process where for each time period the edges in each quarter were randomly rewired, such that the rewiring shuffled the time-stamp of IMs. The results were consistent and statistically unchanged from the reported findings.

Second, we computed the stability ratio for each element of the observed transition matrix for each time period. To compute the stability ratio, we shuffled the networks in each quarter 10,000 times, each time preserving the degree distribution. We computed the transition matrix for every two subsequent randomized networks. Then, we computed the zscore on every element of the network (triad i to triad j transition) and compared each element in the observed network to the corresponding element in the randomized networks, to determine whether the observed transition matrix was statistically significant from the randomized matrices. We report the findings in Table 2 column 3. We found that more than 83% of elements in the observed transition matrices in the manuscript were statistically significant. Table 2 shows the ratio of nonzero statistical significant elements in the transition matrix for each period compared to 10,000 transitions from randomized networks (at the $p < 0.05$ level of significance).

Table 2: Stability ratio of transitions in each transition matrix with respect to 10,000 transition matrices from randomized networks within 95% confidence interval. All transition matrices cannot be explained by chance.

Transition	Stability Ratio
Quarter 1 to Quarter 2	0.84
Quarter 2 to Quarter 3	0.92
Quarter 3 to Quarter 4	0.94
Quarter 4 to Quarter 5	0.91
Quarter 5 to Quarter 6	0.83

Please see column 3 in the Table 2 in the manuscript and more details in page 2, line 90.

- (v) “was the stationary distribution referenced in Table 2 the same for every row (i.e. the stationary distribution of the 5-step Markov chain) shown in table 3, or was a stationary distribution found for each of the 5 transition matrices? Can you walk through the meaning of the distribution given in table 3?”

Table 2 computes the L2-norm distance of stationary probability distribution for each triad from the average stationary distribution by quarter. The stationary distribution given in Table 3 is computed on the average matrix of all 5 transition matrices.

[R2: 3] “Specific notes:”

- (i) “Fig 3. Suitably randomized networks are not clear. What procedure?”

Please refer to [R2: 1c] for more explanation about the randomization algorithm.

- (ii) “Fig 4. Please place error bars on the random network points. (Or is that what the lines denote?). Connecting the 6 quarters with a line is misleading, perhaps consider a bar-chart to indicate all quarters were randomized independently.”

As expected, the lines denote error bar for 10,000 randomized networks. Based on your comment, we removed lines and presented 95% confidence interval error bars for the randomized networks in Fig. 3, and updated the manuscript accordingly.

Figure 3: Comparison of the observed balance in the system, b_t to the expected \hat{b}_t in each time period indicates that the observed system is in a greater state of balance than would be expected in a comparable randomized network.

- (iii) “Did the regression analysis control for IM frequency? If the number of IM’s is related to performance, and IM’s tend to obey structural balance, then one might expect to see a relationship between structural balance and performance.”

We have now controlled for the number of IM’s in the regression. Please see the paper, Figure 6a and Table 4, for the updated coefficient plot.

- (iv) “Fig 6 C (note, there is no b) is very awkward. Only 3 points are shown, and the x-scale is not clear. The y-labels do not clearly show the decimals.”

We have updated Figure 6b (formerly 6 C; thank you for making this point) to include more data points. The x axis is the balance ratio (defined as the # of 102 and 300 triads / all possible triads) and the y axis is the probability of achieving the ”hot hand”. The numbers in the margins plot are inclusive of all controls.

- (v) “There are several grammar mistakes and typos, including pg 6, para 4,”

Thanks. We have fixed these errors.

- (vi) “avoid double negative: not dissimilar, page 6, para 5”

Thanks. We have changed the wording in the text.

Comments by Referee # 3

The authors investigate how affective relationships among 66 stock traders affect their trading performance, over a period of two years. The interrelation among brokers is built by using the instant messages they exchange daily to gain information relevant to their trading. The traders’ performance is monitored from stock market data and further used to investigate whether balanced relationships make traders to perform better or worse.

The study analyzes the largest longitudinal data assembled so far, as reported by the authors, and this favors a unique analysis of the sentiment network across a large time frame. Structural balance theory (SBT) is used as modelling framework for the process under study. The authors claim to find a tendency for the sentiment network to transition into states of greater balance over time. Furthermore, they conclude that the trader’s degree of structural balance is positively correlated

with the trader's performance.

The findings of the study are interesting to understand how social relationships can affect performances. However, some parts of the paper deserve further clarification and limitations of the study should be better addressed. In some part, results are overstated or not fully discussed in their limits and it is hard to draw a final conclusion on the validity of the study. I further elaborate below on these salient points.

Thank you for your review. We hope our responses to your specific concerns will further clarify the results and show all potentials and limitations of the paper. We believe, the changes that have been made in response to your comments, have certainly served to strengthen the paper.

[R3: 1] “The authors claim that the transition probabilities are relatively stable across transition periods and they use, as support of this statement, the L2-norm distance between the current state and the stationary probability distribution (SPD), see Table 2. Within the whole text, though, there is no mention of how the SPD is obtained, which is a crucial quantity for all the results which come next and, therefore, their interpretation is more difficult.”

In the original manuscript, we stated “. . . The stationary distribution is computed as the normalized left eigenvector corresponding the eigenvalue of 1 of the row-stochastic transition matrix”. More precisely, a stationary distribution of a Markov chain is a probability distribution that remains unchanged as time progresses. It has been mathematically proved that an ergodic, aperiodic and irreducible Markov chains has a unique and absorbing stationary distribution [15].

More formally, assume the distribution of triads in the first period is π_1 . Given the transition matrix M , and assuming it is stable then $\pi_2 = \pi_1 M$, $\pi_3 = \pi_2 M$, and so forth. Thus by taking infinite steps we have:

$$\lim_{s \rightarrow \infty} \pi_s = \lim_{s-1 \rightarrow \infty} \pi_s M,$$

Which this yields:

$$\pi_{lim} = \pi_{lim} M$$

This means limiting distribution is the stationary distribution.

In our experiments on the dataset, we showed that the transition matrices and their corresponding stationary distributions are stable over 6 periods (r -value of 0.94 to 0.97 with $p < 0.05$).

Per this comment, we have added more description and a relevant citation about stationary distribution to the manuscript, in page 7, line 16.

[R3: 2] “Structural balance theory predicts that only the triads 300 and 102 should exist in a sentiment network with no violations of the axioms, as remarked by the authors. From the analysis of the stationary states, the authors conclude, indeed, that the triads 300 and 102 have higher occurrences of balanced states. This is partially true from my examination of the results. Indeed, from Table 3 one reads that the state with the highest SPD is the null state, with SPD = 0.68. According to structural balance theory this state is unbalanced and, therefore, should have a null or very small SPD, from my understanding. The authors comment this fact by stating that the high SPD of this state suggests that the network remains sparse over time, which is true, but to me this also signals a limitation of the SBT in the prediction of the stationary states. Furthermore, from Table 3, one reads that the states 012 and 201 (predicted unbalanced by the SBT) have SPT of 0.02 and 0.04, respectively, which are twice and four times higher than the SPD of the state 300, with SPD = 0.01, which is predicted balanced by SBT. Again, this signal a failure or at least a limitation of SBT in the prediction of the stationary states that it is not only worth mention but, perhaps, deserves some further analysis and explanation.”

- (i) Presence of null triad 003 in stationary distribution: you are accurately pointing out an issue raised by us. Based on classical balance theory, triad 003 is not permitted. Rule 4 depicts “an enemy of an enemy is a friend” which should preclude us from observing the null triad 003. However, the stationary distribution shows that

003 is present in the networks. Therefore, our findings show the limitation of classical balance theory which was actually predicted by Heider himself [11] and then introduced formally by Davis [1]. In particular, Heider writes 'if two negative relations are given, balance can be obtained either when the triad relationship is positive or when it is negative, though there appears to be a preference for the positive alternative.' [11]. Davis et al. were the first to relax the classical balanced theory and introduced the Cluster model such that it allows for triad 003 [1].

- (ii) Presence of other unbalanced states (012 and 201) in stationary distribution: Triads 102 and 201 both have a probability of 0.02 in the stationary distribution but this compares to a probability of 0.27 for the classically balanced states. Our results find evidence that both these unbalanced states, 012 and 201, are transitioning towards either the classically balanced triads or the null triad.

First, examining Fig. 2, we see that both triad configurations have positive transitions to 102 or 300, suggesting that they are moving towards greater balance.

Second, although 012 and 201 have a higher prevalence than 300 in the stationary distribution, the probability of observing 300 in any network is very low. Examining Fig. 3, we see that the presence of triad 300 in our setting is higher than in the randomized network, meaning that we actually see more of the 300 triad configuration than in the randomized network. At the same time, we do see that 201 occurs with significantly lower probability.

As a final note, we do find that 102 occurs with higher probability than the randomized network. So, you are right to suggest that there is some limitation of SBT. We interpret this result to signify that although unbalanced triads are moving towards greater balance, these transitions occur slowly. Thus, forbidden triads remain within our observation period.

We have added these points to the discussion in the manuscript, in page 3, line 25 and in page 6, line 77.

- [R3: 3] "To test whether their results are due to chance, the authors compute the likelihood of each triad from suitably randomized networks, see Fig 3. They find that the actual network has a lower occurrence of null triads and, thus, the high SPD of the state 003, cannot even be explained by random effects in the data. How do the authors interpret this result together with the remarks raised in point b) above about the SPD of state 003?"

Fig. 3 shows that triad 003 is less frequent in the real networks compared to the randomized ones. This means if we have random generated networks, with the same degree distribution, we would have more triad 003 than existing ones. However, this is unrelated to the transitions. The transition probability matrices and the corresponding stationary distributions in Fig. 2 and Table 3 show that transitions are likely to occur from unbalanced triads to triad states 300, 102 and 003 and remain stable. Thus, although there are fewer null 003 state triads than expected in a randomized network, null triads persist and are very stable (given the large probability of self-transitions, as shown in the stationary distribution). Accordingly, these two findings do not contradict each other.

- [R3: 4] "To build the sentiment network and compute the transition probabilities the authors fix a quarterly period (four months) as time-window to build the sentiment network and compute the transition probabilities. The construction of the network strongly depends on the time-intervals employed to partition the whole observation period. In Methods, the authors point out that the analysis of data using monthly and bi-monthly time intervals produced similar results. As the authors describe in the text, the traders open and close several trading position every day and they have high volume of instant messages that exchange every day. For this reason, and due to the nature of the problem, certain processes might happen at a much shorter time scale than a month. I wonder if the authors have tried to analyze the data also with weekly and bi-weekly time intervals and, if so, if they have found differences in the stability of the states and in the conclusion of the study. I believe the effect of the time-interval chose to analyze the data on the result to be a very important issue which should be discussed extensively."

The results in the manuscript are computed for quarters (3 months). We reran the experiments for weekly and bi-weekly time intervals. The distribution of triads, transition matrices, and stationary distributions were similar in

Figure 4: Full transition matrix for quarterly, bi-weekly and weekly periods.

these results except that in bi-weekly and weekly periods, there are more null 003 triads, and transitions to the 003 configuration is higher. This makes sense because we are examining a smaller time interval (i.e., 5 or 10 business days), meaning that traders have fewer opportunities to initiate IM's with each other. On average, traders IM with (2-3) other contacts each week.

Per this comment, we strengthened the manuscript by mentioning that our findings are robust to period length, including weekly or bi-weekly intervals, in page 6, line 116.

[R3: 5] “Fig. 4 shows an overall decrease of the classically balanced ratio of the real network. The authors comment that this decline corresponds to the uncertainty due to the 2008-09 financial crash. The figure shows, though, that the same decline is also a feature of the balance ratio of randomly generated networks. These networks, being randomly created, should not be sensitive to the actual data and, thus, to the financial crash of 2008-09. So, I wonder, how are the random networks generated in this study? Why such decline does effect also these random networks?”

Random networks are generated based on the real (observed) networks such that both have the same number of nodes, same number of edges and in and out-degree distributions. For randomization, we used the validated method by Milo et al. [5]. Please also refer to the response [R2: 1c] to reviewer 2. We improved the manuscript by adding more details about randomization method, in page 7, line 56.

References

- [1] James A Davis. "Clustering and structural balance in graphs". In: *Human relations* 20.2 (1967), pp. 181–187.
- [2] Giuseppe Labianca. "Negative ties in organizational networks". In: *Contemporary perspectives on organizational social networks*. Emerald Group Publishing Limited, 2014, pp. 239–259.
- [3] Giuseppe Labianca, Daniel J Brass, and Barbara Gray. "Social networks and perceptions of intergroup conflict: The role of negative relationships and third parties". In: *Academy of Management journal* 41.1 (1998), pp. 55–67.
- [4] Nancy K Baym, Yan Bing Zhang, and Mei-Chen Lin. "Social interactions across media: Interpersonal communication on the internet, telephone and face-to-face". In: *New Media & Society* 6.3 (2004), pp. 299–318.
- [5] Ron Milo et al. "Network motifs: simple building blocks of complex networks". In: *Science* 298.5594 (2002), pp. 824–827.
- [6] Fritz Heider. "Attitudes and cognitive organization". In: *The Journal of Psychology* 21.1 (1946), pp. 107–112.
- [7] Dorwin Cartwright and Frank Harary. "Structural balance: a generalization of Heider's theory." In: *Psychological review* 63.5 (1956), p. 277.
- [8] Eugene C Johnsen. "Network macrostructure models for the Davis-Leinhardt set of empirical sociomatrices". In: *Social networks* 7.3 (1985), pp. 203–224.
- [9] Jure Leskovec, Daniel Huttenlocher, and Jon Kleinberg. "Signed networks in social media". In: *Proceedings of the SIGCHI conference on human factors in computing systems*. ACM. 2010, pp. 1361–1370.
- [10] Vitaly J Dubrovsky, Sara Kiesler, and Beheruz N Sethna. "The equalization phenomenon: Status effects in computer-mediated and face-to-face decision-making groups". In: *Human-computer interaction* 6.2 (1991), pp. 119–146.
- [11] Fritz Helder. "The psychology of interpersonal relations". In: *New York: Wiley*. (1958), p. 206.
- [12] Aage B Sørensen and Maureen T Hallinan. "A stochastic model for change in group structure". In: *Social Science Research* 5.1 (1976), pp. 43–61.
- [13] Noah E Friedkin. "A formal theory of reflected appraisals in the evolution of power". In: *Administrative Science Quarterly* 56.4 (2011), pp. 501–529.
- [14] Craig M Rawlings and Noah E Friedkin. "The structural balance theory of sentiment networks: Elaboration and test". In: *American Journal of Sociology* 123.2 (2017), pp. 510–548.
- [15] Athanasios Papoulis and S Unnikrishna Pillai. *Probability, random variables, and stochastic processes*. Tata McGraw-Hill Education, 2002.

Reviewers' comments:

Reviewer #1 (Remarks to the Author):

I would like to thank the authors for taking the time to respond to my comments and, in the mean time, to learn me some historical aspects about signed networks. After having read their reply and the updated manuscript, I am now confident that the manuscript is suitable for a publication in Nat Comm.

Reviewer #2 (Remarks to the Author):

The authors have made substantial improvements to the validation and clarity of their work. Therefore, I recommend to accept the paper for publication in Nature Communications and applaud the authors for the insights they managed to wrestle from the data.

Reviewer #3 (Remarks to the Author):

The revision of the paper has improved the manuscript, the authors took into account my comments and tried to address all of them. Although I am very satisfied by most of their answers there are few points that I would like them to further address.

First of all, it seems to me that the values of the stationary probability distribution shown in Table 3 have changed compared to the previous version of the manuscript, if I am not mistaken. Have the authors re-run the all analysis and obtained different results for this probability distribution?

[R3:2] The authors cite Heider [38] to discuss that the presence of the triad 003 was predicted in his seminal work and Davis [39] as a reference for a model which allows for the existence of such triad in structural balance. These predictions and model find empirical support in their longitudinal analysis. Yet, according to my understanding, the authors do not make use of the model introduced by Davis and therefore, in their theory, the high value of the stationary probability distribution of the triad 003 (now = 0.75 in Table 3) still results off from the prediction of standard structural balance. One of the authors' conclusion of the study is that is that balanced ties may offer more social support. This claim is certainly supported by the existence of the triad 300 and 102 but it seems in contrast with the high value of the stationary probability distribution of the 003. Indeed, if the presence of this triad is a systemic feature of the data, it would signal that sparsity in the network, i.e. a major tendency into individualism or not communication with the peers, is the most balanced state of the system under investigation. This would significantly change the conclusion of the study. I believe these limitations of the study are worth of further mention and discussion.

[R3:2](ii) The prevalence of the triad 012 and 201 in comparison with the prevalence of the triad 300 has nothing to do with neither the transition probability nor the stationary distributions, as indeed the authors themselves observe in reply to my inquiry in point [R3:3].

I really appreciate that the authors performed the analysis for weekly and bi-weekly time intervals. These new findings (already mentioned in the previous version of the manuscript but not shown) are a great addition to the study. I think these results are worth to be shown in the manuscript together with the comments by the authors in reply to one of my remark (see [R3:4]).

Statement of Revision

Structural Balance Emerges and Explains Performance in Risky Decision-Making

We would like to thank the reviewers for reviewing our paper. We appreciate the thoughtful and constructive comments from all reviewers. In the following, we provide a detailed account of all the changes that we have made in the *second* revised version of the paper. We have structured this list in separate blocks, corresponding to the comments made by the referees. The referee's text in is blue followed by our response is in black. Also, in the revised manuscript all the changes resulting from the revision are in red followed by their page and line number in the main manuscript. We would like to thank all the reviewers for their thoughtful and constructive remarks.

Comments by Referee # 1

I would like to thank the authors for taking the time to respond to my comments and, in the mean time, to learn me some historical aspects about signed networks. After having read their reply and the updated manuscript, I am now confident that the manuscript is suitable for a publication in Nat Comm.

Thank you for your review and recommendation to accept our paper.

Comments by Referee # 2

The authors have made substantial improvements to the validation and clarity of their work. Therefore, I recommend to accept the paper for publication in Nature Communications and applaud the authors for the insights they managed to wrestle from the data.

Thank you for your review and recommendation to accept our paper.

Comments by Referee # 3

The revision of the paper has improved the manuscript, the authors took into account my comments and tried to address all of them. Although I am very satisfied by most of their answers there are few points that I would like them to further address.

Thank you for your review. Below we address your concerns point-by-point.

[R3: 1] "First of all, it seems to me that the values of the stationary probability distribution shown in Table 3 have changed compared to the previous version of the manuscript, if I am not mistaken. Have the authors re-run the all analysis and obtained different results for this probability distribution?"

Table in the first submission (Table 2) was the stationary distribution only for the first transition matrix. In the revision process, we followed the reviewers' recommendations and updated the table to include all transitions for the entire observation period. To include all transitions, we computed the stationary distribution of the average of all transition matrices and added "average Markov chain over all periods" to the table caption (Table 1). Consistent with the expectations of the reviewers that the expanded sample of transitions should not be markedly different from the original sample, the new table shows that the stationary distribution values are not significantly different from the original table. For the sake of comparison, we show both tables in the following,

Table 1: (current table) Stationary distribution of the average Markov chain over all periods. The stationarity of the null triad state suggests that forbidden triads remain in the network.

Triad Type	300	102	003	120D	120U	030T	021D	021U	012	021C	111U	111D	030C	201	120C	210
Stationary Probability	0.02	0.20	0.75	0.00	0.00	0.00	0.00	0.00	0.02	0.00	0.00	0.00	0.00	0.01	0.00	0.00

Table 2: (previous table) Stationary distribution of the Markov chain in the first period. The stationarity of the null triad state suggests that forbidden triads remain in the network.

Triad Type	300	102	003	120D	120U	030T	021D	021U	012	021C	111U	111D	030C	201	120C	210
Stationary Probability	0.01	0.24	0.68	0.00	0.00	0.00	0.00	0.00	0.04	0.00	0.00	0.00	0.00	0.02	0.00	0.00

[R3: 2] “[R3:2] The authors cite Heider [38] to discuss that the presence of the triad 003 was predicted in his seminal work and Davis [39] as a reference for a model which allows for the existence of such triad in structural balance. These predictions and model find empirical support in their longitudinal analysis. Yet, according to my understanding, the authors do not make use of the model introduced by Davis and therefore, in their theory, the high value of the stationary probability distribution of the triad 003 (now = 0.75 in Table 3) still results off from the prediction of standard structural balance.”

We agree that the classical structural balance theory literature clearly expects a stationary distribution for triads 300 and 102 and provides support for that expectation in cross-section data [1, 2, 3, 4] – something we found in our longitudinal study. In our longitudinal study, we also found that triad 003 behaves like triads 300 and 102, an apparently unexpected result. However, on further review of the literature we discovered that Heider’s original conceptual prediction [5], and Davis’ formal model [6] theoretically support the new empirical finding about triad 003. Specifically, however, to your query about using Davis’ model, we cannot directly implement of Davis’ model because of its formal nature. Consequently, we use the reasoning and formal deductions of the model to interpret the meaning of our empirical results.

In particular, Heider writes “if two negative relations are given, balance can be obtained either when the triad relationship is positive or when it is negative, though there appears to be a preference for the positive alternative.” [5]. Davis et al. were the first to relax the classical balance theory and introduced the Cluster model such that it allows for triad 003 [6].

Thus, based on your comment, we clarified that our reference to Davis’ model is not in implementing his formal model but in using its theoretical deductions (in conjunction with Heider’s) to interpret our empirical results. This revision was added (see page 3, lines 18-19)(also page 5, lines 37-39).

[R3: 3] “One of the authors’ conclusion of the study is that is that balanced ties may offer more social support. This claim is certainly supported by the existence of the triad 300 and 102 but it seems in contrast with the high value of the stationary probability distribution of the 003. Indeed, if the presence of this triad is a systemic feature of the data, it would signal that sparsity in the network, i.e. a major tendency into individualism or not communication with the peers, is the most balanced state of the system under investigation. This would significantly change the conclusion of the study. I believe these limitations of the study are worth of further mention and discussion.”

Your observation raises an issue that we reasoned through carefully. While in theory the “003” finding may signal that relational sparsity/individualism typifies balance, we believe that three features of the context of a trading firm, as well as empirical tests, support our conclusions about social support in this setting. The reasons are:

- **The information needs of successful trading decisions works against individualism.** Traders and the firm thrive by making good decisions in a complex information environment. Because market information can move very fast and be scattered widely across sources (social media, news, company reports, talk on the street), a broad and diverse set of connections, rather than individualism, promote a trader’s ability to monitor, identify, and interpret volumes of information they could not monitor on their own – in turn boosting their performance.
- **The emotive nature of trading relationships works again individualism.** Because trading is a highly interdependence process, traders are sensitive to the perceived trustworthiness and social support offered by a connection, which is learned and reinforced through personal interactions [7, 8]. (i) and (ii) would therefore imply that each trader’s performance would depend on including connections of positive sociability and excluding connections of negative sociability. The process of excluding potential connections would produce

network sparseness. To this point, for example, we find that empirical increases in balance are positively associated with performance (i.e., the hot hand).

- **The size of the firm.** Given that the firm has ~60 traders, there is no lack of opportunity or motivation for each trader to form connections with every other trader; instead the sparsity of links are likely due to an active process of avoiding certain connections.
- **Size of observation period.** Our observation period is over a 1.5 years that is adequate time for individuals to “test out” relationships with one another.
- **E-communication coverage.** In this context, e-communications are likely to capture face-to-face interactions because traders can use their computers to observe the market and communicate with social and professional contacts at the same time.
- **Null Models.** We compared the presence of the null (003) triad in the observed network to 10,000 randomized network. Fig 3 in the manuscript, indicates that the null triad occurs with less frequency in the observed network than expected. If individualism was a normal state of affairs, the null would have shown that it is typical (i.e., equivalent to what is expected by chance) rather than atypical (i.e., less than expected by chance).

Hence, based on your comment, we added a concise discussion of the mentioned reasons in the revised manuscript (see page 6, lines 18-22).

[R3: 4] “[R3:2](ii) The prevalence of the triad 012 and 201 in comparison with the prevalence of the triad 300 has nothing to do with neither the transition probability nor the stationary distributions, as indeed the authors themselves observe in reply to my inquiry in point [R3:3].”

That is correct. The prevalence of triads compared to the randomized networks is unrelated to the transition probabilities. We clarified this point in the manuscript (see page 4, line 11-13).

The reason for prevalence of triad 012 and 201 is interpreted as a limitation of our observation periods (see page 4, lines 40-43). Although unbalanced triads are moving towards greater balance, these transitions occur slowly; hence, few forbidden triads remain within our observation period. That said, when we look at all periods, we find the majority of triads are balanced configurations, the stationary probability of being in balanced triads by Davis et al. [6] is ~0.97, and consequently the probability of other unbalanced triads (including 012 and 201) occurring is extremely low. We further described this point (see page 3, lines 16-18). Also when we change period length to weekly or biweekly, we consistently see high probabilities for only the balanced triads.

[R3: 5] “I really appreciate that the authors performed the analysis for weekly and bi-weekly time intervals. These new findings (already mentioned in the previous version of the manuscript but not shown) are a great addition to the study. I think these results are worth to be shown in the manuscript together with the comments by the authors in reply to one of my remark (see [R3:4]).”

We agree with you. We have strengthened the paper by adding the weekly and biweekly result figure to the manuscript as Figure 7 and our reply to your earlier comment (see in page 7, lines 12-20).

References

- [1] Fritz Heider. “Attitudes and cognitive organization”. In: *The Journal of Psychology* 21.1 (1946), pp. 107–112.
- [2] Dorwin Cartwright and Frank Harary. “Structural balance: a generalization of Heider’s theory.” In: *Psychological review* 63.5 (1956), p. 277.
- [3] Eugene C Johnsen. “Network macrostructure models for the Davis-Leinhardt set of empirical sociomatrices”. In: *Social networks* 7.3 (1985), pp. 203–224.
- [4] Craig M Rawlings and Noah E Friedkin. “The structural balance theory of sentiment networks: Elaboration and test”. In: *American Journal of Sociology* 123.2 (2017), pp. 510–548.
- [5] Fritz Heider. “The psychology of interpersonal relations”. In: *New York: Wiley*. (1958), p. 206.
- [6] James A Davis. “Clustering and structural balance in graphs”. In: *Human relations* 20.2 (1967), pp. 181–187.
- [7] Yang Yang, Nitesh V Chawla, and Brian Uzzi. “A network’s gender composition and communication pattern predict women’s leadership success”. In: *Proceedings of the National Academy of Sciences* 116.6 (2019), pp. 2033–2038.
- [8] Bin Liu, Ramesh Govindan, and Brian Uzzi. “Do emotions expressed online correlate with actual changes in decision-making?: The case of stock day traders”. In: *PloS one* 11.1 (2016), e0144945.

****REVIEWERS' COMMENTS:**

Reviewer #3 (Remarks to the Author):

The authors have taken into account my previous comments and remarks and have made substantial changes in the manuscript which, overall, have improved the presentation of their study.

The further comments added about the presence of the triad 003, as well as Fig. 7, are an important addition to the study which also clarify the limits of structural balance theory.

I recommend the paper for publication in Nature Communication.

Statement of Revision

Structural Balance Emerges and Explains Performance in Risky Decision-Making

We would like to thank the reviewers for reviewing our paper. We appreciate the thoughtful and constructive comments from all reviewers. In the following, we provide a detailed account of all the changes that we have made in the *third* revised version of the paper. We have structured this list in separate blocks, corresponding to the comments made by the referees. The referee's text is in blue followed by our response is in black. Also, in the revised manuscript all the changes resulting from the revision are in red followed by their page and line number in the main manuscript. We would like to thank the respectful editor and all the reviewers for their thoughtful and constructive remarks.

Comments by Referee # 3

The authors have taken into account my previous comments and remarks and have made substantial changes in the manuscript which, overall, have improved the presentation of their study. The further comments added about the presence of the triad 003, as well as Fig. 7, are an important addition to the study which also clarify the limits of structural balance theory. I recommend the paper for publication in Nature Communication.

Thank you for your review and recommendation to accept our paper.